# CCLRec: Consensus-driven Contrastive Learning for LLM-enhanced Graph Recommendation

Ting Guo [1]  Dongyu Pei [1]  Litiao Qiu [* 2 3]  Xiaoying Liao [4 5]  Ke Liang [* 6]  Peng Song [7]
Pinle Qin [1]

## Abstract

Recommendation systems seek to accurately model user preferences from a large set of candidate items. Graph neural networks (GNNs) have emerged as a dominant approach in this domain due to their ability to capture high-order user–item interactions. Recent efforts have aimed to enhance GNN-based representation learning by incorporating the semantic reasoning capabilities of large language models (LLMs). However, existing methods often process graph structural information and LLM-derived semantic knowledge separately, creating a supervisory gap between structural proximity and semantic relevance. To bridge this gap, we propose CCLRec, a consensus-driven contrastive learning framework for recommendation. CCLRec deeply integrates structural and semantic information by identifying consistent signals. Specifically, we first use an LLM to extract semantic representations of items and to sample candidate positive/negative sets in the semantic space. We then introduce a structural–semantic consensus mining strategy that computes the intersection between a node's structural neighbors in the graph and its semantically similar items. This allows us to identify high-confidence positive pairs endorsed by both collaborative filtering patterns and LLM-based reasoning. By centering contrastive learning on these consensus pairs and applying a weight-aware reinforcement mechanism during training, CCLRec significantly amplifies the contribution of high-quality consensus features during training. Experiments across multiple public benchmarks show that CCLRec consistently outperforms state-of-the-art methods on key metrics, demonstrating the effectiveness of our consensus-aware design.

## 1. Introduction

In the era of information explosion, recommendation systems serve as a critical bridge connecting users with content of potential interest, playing an indispensable role in mitigating information overload and enhancing user experience (Meng et al., 2023; Wang et al., 2015; Zhang et al., 2020; Shi et al., 2012). Within this field, graph neural networks (GNNs) have established themselves as a dominant paradigm, owing to their powerful capacity for topological modeling (Liang et al., 2024; Guo et al., 2024). By constructing user-item interaction histories as bipartite graphs, GNNs leverage message-passing mechanisms to aggregate information from high-order neighbors, thereby explicitly encoding complex collaborative signals in the embedding space (Guo et al., 2020; Gao et al., 2023b). However, this collaborative filtering paradigm is inherently constrained by the topological properties of the interaction graph. On the one hand, in the face of prevalent interaction sparsity and long-tail distributions in real-world scenarios, graph-based propagation mechanisms tend to fail due to weak signals. On the other hand, interaction graphs often contain a substantial number of noisy edges caused by randomness, which directly interferes with the modeling of genuine user preferences. In the above cases, the graph-based recommendation models often struggle to learn high-quality representations due to insufficient topological support, leading to significant degradation in recommendation performance.

To overcome these structural limitations, research focus has gradually shifted towards leveraging auxiliary modal information (e.g., semantic content, multi-modal information), to enhance representation learning and gain a deeper understanding of user intent beyond mere interaction patterns (Chen et al., 2025; Wei et al., 2024). In recent years, the

---

[1]Department of Computer Science and Technology, North University of China, Shanxi, China [2]Kylinsoft Company Limited, Tianjin, China [3]Harbin Institute of Technology, Harbin, China [4]Changsha Bus Group, Changsha,China [5]Central South University of Forestry and Technology, Changsha, China [6]National University of Defense Technology, Changsha, 410000, China [7]Shanxi University, Shanxi, China. Correspondence to: Litiao Qiu <qiulitiao@kylinos.cn; 24B903134@stu.hit.edu.cn>, KE LIANG <liangke200694@gmail.com>.

*Proceedings of the $43^{rd}$ International Conference on Machine Learning*, Seoul, South Korea. PMLR 306, 2026. Copyright 2026 by the author(s).

advent of large language models (LLMs) has introduced transformative potential for recommendation systems (Zhao et al., 2024; Behdin et al., 2025; Kim et al., 2024). Equipped with extensive world knowledge and superior logical reasoning capabilities acquired from pre-training on massive corpora, LLMs can deeply comprehend the textual semantics of items and infer latent user preferences from historical behavior sequences. Integrating LLMs into recommendation systems not only enriches items with deep semantic representations that transcend the limitations of pure interaction structures but also enables knowledge transfer through semantic correlations in data-sparse scenarios. The deep integration of structural information with semantic knowledge opens new avenues for addressing sparsity challenges and enhancing the interpretability and generalization capability of recommendation systems.

However, existing LLM-enhanced graph recommendation methods often face a fundamental challenge: the conflict between structural and semantic information. Specifically, prior approaches typically treat collaborative signals reflected by graph structures and semantic knowledge derived from LLMs as two separate or loosely coupled views, optimizing them independently or applying simple fusion. Such a paradigm struggles to reconcile topological proximity (items co-occur in user interaction histories) and semantic relevance (items share similar content meanings). When structural neighborhoods diverge from semantic neighborhoods, the model receives conflicting supervisory signals, which not only limits further improvements in representation quality but may also introduce additional noise.

To address these challenges, we propose a novel consensus-driven contrastive learning framework for LLM-enhanced graph recommendation, CCLRec. The core idea is to move beyond treating structural and semantic views as independent sources of supervision. We proactively explore and reinforce their consistency as a foundation for high-quality, high-confidence supervisory signals. Concretely, we first leverage an LLM to generate rich semantic representations for items and users, identifying potential positive and negative samples in the semantic space. We further introduce a structure–semantics collaborative consensus mining strategy. This approach extracts high-confidence positive and negative samples by computing the intersection of structural and semantic views, ensuring dual validation from collaborative signals and LLM reasoning. This mechanism fundamentally alleviates the supervisory conflicts commonly encountered in contrastive learning when structural and semantic views are optimized independently. Building upon the mined consensus pairs, we design a consensus-driven contrastive learning objective and incorporate a weighted mechanism to adaptively amplify the contribution of high-quality consensus features during training. In summary, our main contributions are as follows:

- We propose CCLRec, a consensus-driven contrastive learning framework for recommendation that deeply integrates graph structural and LLM semantics information.

- We design a structure–semantics collaborative consensus mining strategy to identify cross-view consistent sample pairs, providing high-confidence supervision for contrastive learning and effectively mitigating supervisory conflicts in multi-view learning.

- A consensus-driven contrastive loss is constructed that enables the model to reinforce the effect of consensus samples pairs, promoting mutual enhancement between structural proximity and semantic relevance.

- We conduct extensive experiments on multiple public benchmark datasets. Results demonstrate that CCLRec significantly outperforms state-of-the-art methods across key evaluation metrics.

## 2. Related Work

### 2.1. Graph-based Recommendation

Graph-based recommendation methods model user–item interactions as graphs to capture high-order collaborative relationships beyond matrix factorization (Wu et al., 2022; Gao et al., 2023a). Representative GNN-based collaborative filtering methods, such as NGCF (Wang et al., 2019) and LightGCN (He et al., 2020), leverage neighborhood aggregation to model high-order dependencies. Graph-based hybrid methods, including GHRS (Darban & Valipour, 2022), incorporate multiple node and relation types to enhance preference propagation. Graph structure learning approaches, such as GSLRRec (Sang et al., 2024), further improve robustness by jointly optimizing graph structures and representations. In addition, multimodal graph-based methods, including LATTICE (Zhang et al., 2021), MICRO (Zhang et al., 2022), and MMSSL (Wei et al., 2023), exploit modality-aware item–item relations to enrich representations. Overall, graph-based recommendation provides an effective framework for modeling complex user-item interactions and capturing high-order collaborative signals.

### 2.2. Semantic-aware Recommendation

Semantic-aware recommendation methods incorporate knowledge graphs or semantic item information to enhance user-item interactions and alleviate data sparsity. AKGE and ATBRG (Sha et al., 2021; Feng et al., 2020) construct semantic neighborhoods based on user–item related subgraphs, emphasizing informative associations and efficient information propagation. CKE integrates knowledge graph embeddings with textual and visual features to enrich item semantics. SKGRec (Xu et al., 2025) fuses semantic fea-

tures with knowledge graph relations to improve entity representations and recommendation quality. In addition, TASTE (Liu et al., 2023) and UniTRec (Mao et al., 2023) leverage textual signals to enhance sequential and text-based recommendations. Overall, these methods highlight the importance of semantic information in improving representation learning and recommendation performance.

### 2.3. Large Language Model–Driven Recommendation and Data Augmentation

Large language models (LLMs) have recently been introduced into recommender systems to address data sparsity and cold-start issues through their strong semantic and generative capabilities (Zhao et al., 2024). Some studies employ LLMs for data augmentation by generating pseudo-interactions or enriching item attributes, as exemplified by LLMRec and LLMSeR (Wei et al., 2024; Sun et al., 2025). Other works align LLM-based semantic representations with collaborative signals to support recommendation generation or downstream tasks such as retrieval and ranking, including TALLRec, LC-Rec (Bao et al., 2023; Zheng et al., 2024), and CORONA (Chen et al., 2025), which proposes a coarse-to-fine framework for graph-based recommendation with LLMs. Moreover, several approaches integrate LLMs with knowledge graphs to enhance graph-based recommendation. Overall, LLM-driven methods offer new opportunities for recommendation, while effective integration with collaborative signals and graph-based structures remains an open challenge.

## 3. Methodology

### 3.1. Problem Statement

This work focuses on graph-based recommendation tasks that integrate semantic and structural information. Let the user set be denoted as $\mathcal{U}$ and the item set as $\mathcal{V}$. The interaction graph is defined as $Gr = (\mathcal{U}, \mathcal{V}, A)$, where $A_{uv} = 1$ indicates that user $u$ has interacted with item $v$, and $A_{uv} = 0$ otherwise. For each user $u$, $\mathcal{H}_u$ denotes the sequence of items in the user's historical interactions. The user's textual representation $P_u$ is generated by a large language model based on the user's historical interactions. While the item's textual representation $T_v$ corresponds to the item's own description. $e_u$ and $e_v$ are embeddings of users and items, respectively. Followed LLMREC, item representations are obtained via element-wise additive multimodal fusion: the ID embedding derived from graph convolution serves as the backbone, onto which image and text features are projected with weights and aggregated to yield the final fused embedding. For user embedding, historical item attributes undergo knowledge completion and semantic augmentation via a LLM, followed by encoding through a text encoder to produce dense embeddings. Further implementation details

are provided in Appendix A. Given the interaction graph $Gr$, the objective is to perform Top-$K$ recommendation, i.e., to predict the next item $v \in \mathcal{V}$ that user $u$ is most likely to interact with.

### 3.2. Framework Overview

Figure 1 depicts the overall workflow of the proposed CCLRec framework. CCLRec operates on a user-item interaction graph and is designed to reconcile structural signals and semantic information. The framework begins by constructing an item co-occurrence matrix from interaction data to capture collaborative relationships. Concurrently, an LLM is employed to perform semantic modeling: it encodes item attributes and user historical behaviors, deriving user interest representations from the semantic embeddings of their interacted items. The detailed procedure is provided in the Appendix A. **At the semantic level**, the model computes the similarity between candidate items and the user's historical item embeddings, selecting the Top-$s$ most similar items as the **semantic positive set**. Items with similarity below a predefined threshold are randomly sampled to form the **semantic negative set**. **At the structural level**, item relevance is computed via row-wise aggregation over the co-occurrence matrix. The Top-$s$ items with the highest co-occurrence scores form the **structural positive set**. An item is deemed eligible for the **structural negative set** only when its co-occurrence with every structural positive sample remains below the designated threshold. Finally, samples that are selected as positive (or negative) in both the semantic and structural views are identified as **consensus samples**. These high-confidence consensus pairs are emphasized during contrastive learning, thereby mitigating supervisory conflicts inherent in single-view learning and enabling a deeper, more synergistic integration of semantic relevance and topological proximity.

### 3.3. Structure-Semantic Consensus-driven Sampling

In graph-based recommendation systems, user preferences are intrinsically driven by both collaborative filtering signals and item semantic information. However, relying on a single source is prone to noisy supervision: structure-based sampling can amplify spurious co-occurrences in sparse data, while semantic similarity may overlook actual interaction patterns, yielding behaviorally irrelevant results. To address this, we propose a structure–semantic collaborative consensus mining strategy. We model user–item relationships independently within the interaction graph and the semantic embedding space, constructing sample sets under dual constraints of structural consistency and semantic affinity. By computing the intersection of graph-structural neighbors and semantically similar candidates, we accurately isolate high-confidence positive samples that are mutually corroborated by collaborative signals and LLM-enhanced reasoning.

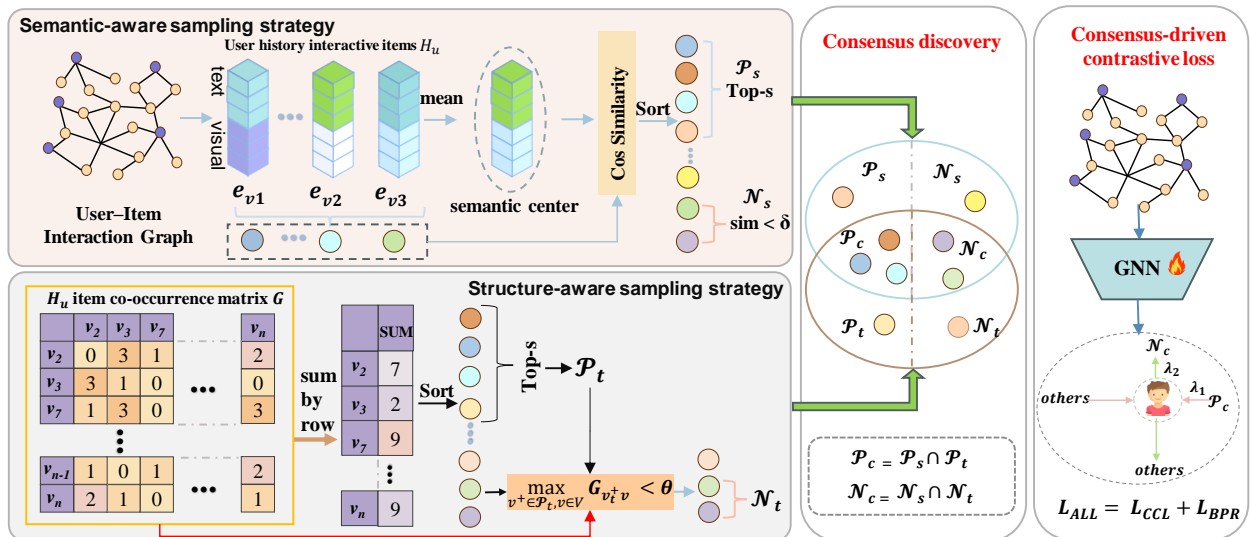

*Figure 1.* The overall framework of our proposed CCLRec.

### 3.3.1. SEMANTIC-AWARE SAMPLING STRATEGY

Although structural signals directly originate from actual user interactions and are generally reliable, they are inherently constrained by historical behavior data, making it difficult to cover latent user interests that have not been explicitly expressed. In sparse interaction or diverse-interest scenarios, relying solely on structural consistency may lead to overly conservative modeling and limit the generalization capability of the model. To address this limitation and fully leverage the reasoning ability of LLMs, we propose a **semantic-aware sampling strategy**. This strategy utilizes pre-trained LLM semantic embeddings and identifies potential user semantic interests through vectorized global similarity computation, while establishing clear semantic discrimination boundaries.

**Semantic positive mining from historical items**. To identify the most semantically representative items within a user's history, we compute a semantic centroid by performing mean-pooling over all historical item embeddings. The user's semantic profile $a_u$ is formulated as:

$$a_u = \text{Normalize}\left(\frac{1}{|\mathcal{H}_u|}\sum_{v \in \mathcal{H}_u} e_v\right), \quad (1)$$

where $\mathcal{H}_u$ denotes the user's interaction set. We then measure the semantic affinity between each historical item $v \in \mathcal{H}_u$ and this centroid using cosine similarity:

$$s_v = \text{sim}(e_v, a_u), \quad (2)$$

where sim denotes the similarity function. Finally, the Top-$s$ items with the highest scores $s_v$ are selected to form the semantic positive set $\mathcal{P}_s$, capturing the items that are central to the user's core semantic preferences.

**Semantic negative sample mining**. To construct negative samples that are semantically distinct from the user's interests, we employ a threshold-based rejection sampling strategy. First, we strictly exclude all items in the user's interaction history $\mathcal{H}_u$ from the candidate pool to avoid false negatives. Subsequently, we randomly sample candidates from the remaining item set. A candidate item $v$ is retained in the semantic negative set $\mathcal{N}_s$ if and only if its cosine similarity with the user's semantic centroid $a_u$ falls below a predefined threshold $\delta$ (i.e., $s_v < \delta$), where $v \in \mathcal{V}$, $s_v = \text{sim}(e_v, a_u)$. This constraint ensures that the selected negatives maintain a sufficient semantic margin from the user's historical preference profile.

### 3.3.2. STRUCTURE-AWARE SAMPLING STRATEGY

To accurately capture high-order user-item interactions, traditional random sampling often proves inadequate, as it frequently introduces substantial noise and neglects the topological characteristics of graph data. While the majority of existing graph-based recommendation methods employ random node dropout or edge masking to generate augmented views, these stochastic strategies suffer from two critical limitations: (1) they may inadvertently sever crucial connectivity within the user interest graph, leading to degraded representation quality; and (2) they fail to differentiate between interaction noise (e.g., accidental clicks) and core preferences. To address these issues, we propose a structure-aware sampling strategy. By constructing an item co-occurrence matrix from the interaction graph, this method explicitly identifies positive item samples that embody the user's core interests, as well as negative samples that possess clear discriminative boundaries.

**Item co-occurrence matrix construction**. Two items that

are interacted with by many users are likely to share strong intrinsic correlations. Given the original binary user–item interaction matrix $\mathbf{R} \in \{0,1\}^{|\mathcal{U}| \times |\mathcal{V}|}$, we compute the second-order connectivity between items: $\mathbf{G} = \mathbf{R}^\top \mathbf{R}$, where $\mathbf{G}$ denotes the item–item co-occurrence matrix. Each element records the number of users who have interacted with same items. We set the diagonal elements to zero.

**Structured positive sample mining**. We derive a submatrix $\mathbf{G}' \in \mathbf{G}$ indexed by the items in $\mathcal{H}_u$, which encapsulates the mutual structural support among the user's behaviors. The structural importance score of an item $v_i \in \mathcal{H}_u$ is then calculated by aggregating its connections within $\mathbf{G}'$:

$$\text{Score}_{\text{struct}}(v_i) = \sum_{v_j \in H_u, v_i \neq v_j} G'_{v_i v_j}. \tag{3}$$

The score reflects the overall co-occurrence strength between item $v$ and the other items in $\mathcal{H}_u$. A higher structural support indicates that the item receives broader collaborative support, and is more likely to represent the user's stable preference pattern. We select the Top-$s$ items with the highest scores to form the structured positive sample set $\mathcal{P}_{\text{t}}$.

**Structured negative sample mining**. To accurately select negative samples that are truly structurally irrelevant to a user's core interests, we propose a structure-aware negative sampling strategy based on item co-occurrence relations. This approach ensures that sampled negatives remain sufficiently distinct from user preferences at the structural level. For any candidate item $v \in \mathcal{V}$, structural relevance is determined by its co-occurrence strength with the user's positive samples. We identify $v$ as a valid candidate for structured negative sampling if its maximum co-occurrence with all positive samples falls below a threshold:

$$\max_{v^+ \in \mathcal{P}_{\text{t}}, v \in V} \mathbf{G}_{v_t^+ v} < \theta, \tag{4}$$

where items satisfying this condition are regarded as structurally irrelevant. Consequently, the structured negative sample set $\mathcal{N}_t$ is sampled exclusively from items with a mask value of 0. This set consists of items that have not been interacted with by the user and are structurally irrelevant to their core interests. This selection criterion implies that item $v$ is topologically distant from the user's interest subgraph, thereby minimizing the risk of introducing false negatives (i.e., unobserved items that are actually consistent with user preferences) and ensuring the model learns from high-quality signals.

### 3.4. Loss Function

#### 3.4.1. CONSENSUS-DRIVEN CONTRASTIVE LOSS

To address potential conflicts between structural and semantic signals in LLM-enhanced graph recommendation, while fully leveraging their complementarity, we propose

a consensus-driven contrastive loss. The core idea of this mechanism is to identify samples that are validated by both the graph-structural view and the semantic view, thereby locking high-confidence supervisory signals and weighting their contributions in contrastive learning. For each user $u$, samples are obtained via structure-aware and semantic-aware sampling, respectively. We determine the consensus state through a set intersection operation. Specifically:

$$\mathcal{P}_c = \mathcal{P}_s \cap \mathcal{P}_t, \quad \mathcal{N}_c = \mathcal{N}_s \cap \mathcal{N}_t. \tag{5}$$

If the intersection is non-empty, we consider that the user's preference reaches consensus across the two views. Based on this, we introduce consensus weights, with $\lambda_1$ and $\lambda_2$ denoting the weights for positive and negative samples, respectively. To model the collaborative consensus between structural neighborhood consistency and semantic similarity, we design a contrastive objective that integrates explicitly weighted negatives and in-batch implicit negatives. For an anchor user $u$, the loss function is defined as:

$$\begin{aligned} l_{\text{CCL}}^u = -\lambda_1 \sum_{v,v' \in \mathcal{P}_c} \log \frac{\exp(e_v^\top e_{v'}/\tau)}{Z(v)} \\ - \sum_{v,v' \in (\mathcal{P}_t \cup \mathcal{P}_s) \backslash \mathcal{P}_c} \log \frac{\exp(e_v^\top e_{v'}/\tau)}{Z(v)}, \end{aligned} \tag{6}$$

where the normalization term $Z(v)$ is defined as:

$$\begin{aligned} Z(v) = \lambda_2 \sum_{v,v' \in \mathcal{N}_c} \exp(e_v^\top e_{v'}/\tau) \\ + \sum_{v,v' \in (\mathcal{N}_t \cup \mathcal{N}_s) \backslash \mathcal{N}_c} \exp(e_v^\top e_{v'}/\tau). \end{aligned} \tag{7}$$

Here, $e_v$ represents the final embedding vectors of item. $\tau$ is the temperature parameter. The positive consensus weight $\lambda_1$ is applied externally to the log-likelihood, linearly amplifying the gradient contribution of high-quality positive samples. The negative consensus weight $\lambda_2$ is applied internally to the explicit negative term, strengthening the repulsion and establishing a clear decision boundary. When the consensus set $\mathcal{P}_c = \emptyset$, the weighting term naturally vanishes, and CCLRec smoothly transitions to unweighted dual-view joint supervision over the union set $\mathcal{P}_t \cup \mathcal{P}_s$. This mechanism ensures that the model continues to exploit the complementarity between structural and semantic views even without consensus reinforcement. Empirically, with $s = 2$, the proportions of non-empty consensus samples on Netflix and ML-1M reach 69.3% and 7%, respectively. This demonstrates that even in scenarios characterized by sparsity or significant view divergence, the model successfully captures sufficient anchor signals to prevent view misalignment.

### 3.4.2. RANKING LOSS

To optimize the recommendation ranking performance, we adopt the bayesian personalized ranking (BPR) loss as the objective for the main task. Given a user $u$, a positive item $v^+ \in (\mathcal{P}_t \cup \mathcal{P}_s)$, and a negative item $v^- \in (\mathcal{N}_t \cup \mathcal{N}_s)$, the BPR loss is defined as:

$$l^u_{\text{BPR}} = - \sum_{(u,v^+,v^-)\in\mathcal{O}} \ln \sigma\big(\hat{y}_{uv^+} - \hat{y}_{uv^-}\big) + \|\Theta\|_2^2, \quad (8)$$

where $\mathcal{O}$ denotes the set of pairwise training instances for user $u$, and $\hat{y}_{uv} = e_u^\top e_v$ represents the difference between user $u$'s predicted preference scores, as estimated by the model. $\sigma(\cdot)$ is the sigmoid activation function. Here, $e_u$ and $e_v$ denote the final user and item embeddings that integrate both structural and semantic information.

### 3.4.3. OVERALL OBJECTIVE

The overall training objective consists of two components to train a GNN model: the BPR loss for the main recommendation task, the consensus-driven contrastive loss. The final objective function is formulated as:

$$\mathcal{L}_{\text{ALL}} = \mathcal{L}_{\text{CCL}} + \mathcal{L}_{\text{BPR}} = \sum_{u\in\mathcal{U}} l^u_{\text{CCL}} + \sum_{u\in\mathcal{U}} l^u_{\text{BPR}}, \quad (9)$$

## 4. Experiments

We evaluate the proposed CCLRec by addressing the following four research questions: **RQ1 (Overall Performance)**: How does CCLRec perform compared to state-of-the-art baselines across various benchmark datasets? **RQ2 (Ablation Study)**: How do the key components, specifically semantic-enhanced modeling, item co-occurrence structure modeling, and the consensus-driven contrastive learning mechanism, contribute to the model's overall effectiveness? **RQ3 (Hyperparameter Sensitivity)**: How do critical hyperparameters impact the model's performance? **RQ4 (Model Efficiency)**: How does CCLRec compare against existing methods in terms of computational complexity?

### 4.1. Experimental Settings

#### 4.1.1. DATASETS

To ensure a fair comparison, we directly use the preprocessed and released multimodal dataset Netflix from Wei et al. (2024), which augments items in the Netflix Prize data with crawled movie posters and metadata (e.g., titles, release years, and genres). For the widely used MovieLens-1M (ML-1M) benchmark (Cui et al., 2025), which lacks native visual data, we employed the same multimodal data construction pipeline to acquire corresponding movie posters. The statistics for both datasets are summarized in Appendix B Table 4. For visual representation learning, we encoded

all images using a pre-trained CLIP-ViT (Radford et al., 2021) model to extract fixed-dimensional feature vectors. To augment auxiliary semantic information, we deployed the open-source LLMs, *Qwen-7B*, locally for cost-effective data generation. Specifically, the LLM utilizes its internal world knowledge to impute missing item attributes (e.g., director, region, and language) and infers implicit user interests by processing users' historical interaction sequences.

#### 4.1.2. IMPLEMENTATION DETAILS

All experiments are implemented using PyTorch 2.7.1 with Python 3.11.14 on Ubuntu 20.04, and are accelerated by CUDA 12.8. Experiments are conducted on a server equipped with an NVIDIA RTX 5090 GPU with 32GB of memory. During model training, we adopt a unified set of hyperparameters for both the Netflix and ML-1M datasets. The learning rate is fixed to $10^{-3}$, and the embedding dimensionality is set to 64. For semantic enhancement, *Qwen-7B* is employed as the backbone large language model for user interest modeling and item semantic completion. During the use of LLMs, the temperature parameters for the user and item sides are set to 0.3 and 0.5, respectively. For positive and negative sample construction, the sizes of both semantic and structural candidate sets are set to $s = 3$. The positive and negative sample weights are set to $\lambda_1 = 3$ and $\lambda_2 = 10$, respectively, to balance the influence of positive and negative samples during training. The temperature parameter of the consensus sample contrastive loss is set to $\tau = 0.1$ to control the sharpness of embedding similarity distributions. The semantic threshold is set to $\delta = 0.3$ to guide the selection of semantically dissimilar items for negative sampling. The structural co-occurrence threshold is set to $\theta = 5$, which filters out candidate items that are structurally weakly related to core positive samples. The source code is publicly available at https://github.com/dongyu-pei/CCLRec.

#### 4.1.3. BASELINES AND EVALUATION METRICS

To demonstrate the superiority of CCLRec, we compare it with a comprehensive range of baselines spanning four categories: (1) *Classical collaborative filtering (CF) methods*, including **MF-BPR** (Rendle et al., 2012) and the neural-based **NFM** (He & Chua, 2017); (2) *GNN-based models*, specifically **NGCF** (Wang et al., 2019) and the state-of-the-art **LightGCN** (He et al., 2020); (3) *Side-information enhanced approaches*, covering KG-based methods (**CKE** (Zhang et al., 2016), **LATTICE** (Zhang et al., 2021)) and semi-supervised multi-modal models (**MICRO** (Zhang et al., 2022), **MMSSL** (Wei et al., 2023)); and (4) *LLM-empowered recommenders*, including **LLM-Rec** (Wei et al., 2024) and **CORONA** (Chen et al., 2025). In our experiments, we evaluate the recommendation system using **Recall@10/20** and **NDCG@10/20**, where the

*Table 1.* Performance comparison on two datasets in terms of Recall@10/20 and NDCG@10/20.

| METHOD | NETFLIX | | | | ML-1M | | | |
|---|---|---|---|---|---|---|---|---|
| | R@10 | R@20 | N@10 | N@20 | R@10 | R@20 | N@10 | N@20 |
| MF-BPR | 0.0282 | 0.0542 | 0.0140 | 0.0205 | 0.1257 | 0.2048 | 0.3109 | 0.3062 |
| NGCF | 0.0347 | 0.0699 | 0.0161 | 0.0235 | 0.1373 | 0.2157 | 0.3866 | 0.4450 |
| LIGHTGCN | 0.0352 | 0.0701 | 0.0160 | 0.0238 | 0.1436 | 0.2313 | 0.3605 | 0.3502 |
| NFM | 0.0246 | 0.0368 | 0.0239 | 0.0301 | 0.1346 | 0.2129 | 0.3558 | 0.3379 |
| CKE | 0.0420 | 0.0677 | 0.0297 | 0.0426 | 0.1524 | 0.2373 | 0.3783 | 0.3609 |
| LATTICE | 0.0433 | 0.0737 | 0.0181 | 0.0259 | 0.1599 | 0.2523 | 0.4420 | 0.4961 |
| MICRO | 0.0466 | 0.0764 | 0.0196 | 0.0271 | 0.1560 | 0.2463 | 0.4413 | 0.4943 |
| MMSSL | 0.0455 | 0.0743 | 0.0224 | 0.0287 | 0.1441 | 0.2262 | 0.4251 | 0.4796 |
| LLMREC | 0.0531 | 0.0829 | 0.0272 | 0.0347 | 0.1597 | 0.2515 | 0.4425 | 0.4970 |
| CORONA | 0.0616 | 0.0938 | 0.0279 | 0.0416 | 0.1621 | 0.2565 | 0.4468 | 0.5027 |
| CCLREC (OURS) | **0.0682** | **0.1154** | **0.0347** | **0.0464** | **0.1694** | **0.2639** | **0.4567** | **0.5105** |

*Table 2.* Ablation study of different components on two datasets in terms of Recall@10/20, and NDCG@10/20.

| METHOD | NETFLIX | | | | ML-1M | | | |
|---|---|---|---|---|---|---|---|---|
| | R@10 | R@20 | N@10 | N@20 | R@10 | R@20 | N@10 | N@20 |
| W/O CONTRASTIVE | 0.0558 | 0.0878 | 0.0282 | 0.0361 | 0.1597 | 0.2523 | 0.4450 | 0.4995 |
| W/O SEMANTIC | 0.0655 | 0.0975 | 0.0292 | 0.0377 | 0.1487 | 0.2341 | 0.4240 | 0.4807 |
| W/O STRUCTURE | 0.0645 | 0.1002 | 0.0324 | 0.0413 | 0.1507 | 0.2364 | 0.4275 | 0.4837 |
| W/O CONSENSUS | 0.0639 | 0.1062 | 0.0298 | 0.0395 | 0.1629 | 0.2569 | 0.4483 | 0.5025 |
| W/O BPR | 0.0658 | 0.1094 | 0.0310 | 0.0441 | 0.1053 | 0.1797 | 0.3593 | 0.4126 |
| CCLREC | **0.0682** | **0.1154** | **0.0347** | **0.0464** | **0.1694** | **0.2639** | **0.4567** | **0.5105** |

numbers 10 and 20 indicate the sizes of the Top-$K$ recommendation lists. Recall measures the proportion of a user's true interacted items that appear in the Top-$K$ recommended results, reflecting the coverage ability of the recommendations. NDCG further accounts for the ranking positions of the correctly recommended items within the Top-$K$ results, assigning higher weights to items appearing at higher ranks, and providing a comprehensive assessment of ranking quality. These metrics are widely adopted in recommendation system research, suitable for Top-$K$ recommendation tasks.

### 4.2. Overall Performance (RQ1)

To address **RQ1**, we present the comprehensive evaluation results of CCLRec and all baselines on the Netflix and ML-1M datasets in Table 1. Overall, CCLRec consistently yields the best performance across all metrics on both datasets, validating its effectiveness for Top-$K$ ($K = 10, 20$) recommendation. Specifically, on the high-sparsity Netflix dataset, CCLRec attains a Recall@20 of 0.1154 and an NDCG@20 of 0.0464, significantly outperforming the strongest LLM-enhanced baseline, CORONA (0.0938 / 0.0416), by a substantial margin. Similar superiority is observed on the denser ML-1M dataset, where CCLRec achieves a Recall@20 of 0.2639, surpassing both LLMRec and CORONA. These results indicate that our proposed CCLRec effectively enhances both recommendation coverage and ranking quality.

In terms of baseline categories, traditional CF methods (e.g., MF-BPR, NGCF, and LightGCN) exhibit relatively inferior

performance due to their limited capacity to capture high-level semantic information. While knowledge graph-based and multimodal approaches (e.g., LATTICE and MMSSL) improve upon CF by incorporating side information, they still fall short of LLM-enhanced methods. By integrating consensus-driven contrastive learning, CCLRec establishes a new state-of-the-art compared to existing LLM-based recommendation. Furthermore, analyzing the results from a data perspective reveals the robustness of our model. The Netflix dataset is characterized by high sparsity and semantic uncertainty, making effective semantic modeling critical. CCLRec's pronounced advantage here highlights its ability to alleviate sparsity issues. Conversely, the ML-1M dataset provides denser interaction signals and stable structural patterns. Despite these distinct data characteristics, CCLRec maintains superior performance on both, demonstrating the ability to adapt to varying data densities.

### 4.3. Ablation Study (RQ2)

To answer **RQ2**, we conduct an ablation study to evaluate the contribution of each component in CCLRec, with the results summarized in Table 2. We define the variants as follows: **w/o contrastive** removes both structural and semantic contrastive losses, relying solely on the BPR objective; **w/o semantic** and **w/o structure** discard the contrastive loss from the semantic view and the structural view, respectively; **w/o consensus** retains dual-view contrastive learning but excludes the consensus-based sample selection strategy; and **w/o BPR** eliminates the BPR ranking loss, optimizing the

model solely via the contrastive objective.

As shown in Table 2, CCLRec consistently outperforms all variants, demonstrating that each component is indispensable. We highlight three key observations: (1) Impact of contrastive learning: Removing the contrastive module (w/o contrastive) leads to significant performance degradation, particularly on the sparse Netflix dataset (Recall@20 drops from 11.54 to 8.78). This confirms that contrastive learning provides critical self-supervised signals to alleviate data sparsity. (2) Efficacy of consensus: The performance gap between w/o consensus and CCLRec validates our core hypothesis: Blindly treating all structural/semantic neighbors as positive samples introduces noise. The consensus mechanism effectively filters high-confidence pairs to align the two views. (3) Necessity of ranking loss: While w/o BPR performs reasonably on Netflix, it suffers a drastic collapse on the denser ML-1M dataset (Recall@20 drops to 17.97%). This indicates that while contrastive learning improves representation quality, the BPR objective remains essential for optimizing the precise relative ordering of items.

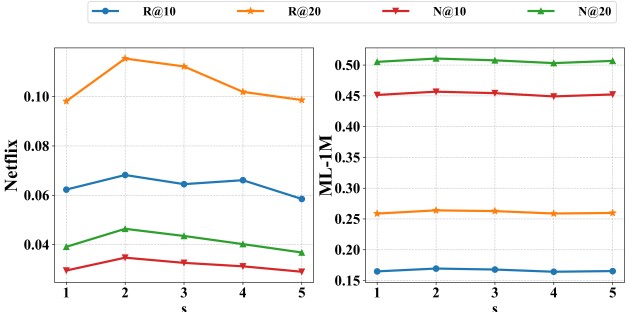

*Figure 2.* The effect of sampling number $s$.

## 4.4. Hyperparameter Sensitivity (RQ3)

**The effect of sampling number $s$.** We evaluate the effect of the sampling number $s$ for positive and negative samples, as this parameter directly affects the adequacy of user-item preference contrast during training. Figure 2 visualizes the impact of different sampling numbers on recommendation performance. Experiments are conducted with $s \in \{1, 2, 3, 4, 5\}$ on both datasets. The results show that the model achieves its best performance when $s = 2$. Specifically, on the Netflix dataset, the model achieves a Recall@20 of 0.1154 and an NDCG@20 of 0.0464. On the ML-1M dataset, it achieves a Recall@20 of 0.2639 and an NDCG@20 of 0.5105. Moreover, $s = 3$ consistently maintains competitive performance, indicating a favorable trade-off between effectiveness and stability.

**The effect of consensus weights $\lambda_1$ and $\lambda_2$.** We investigate the effect of consensus weights on model performance, as these parameters determine the relative influence of positive and negative feedback signals in the training objective. Figure 3 presents the performance variation across differ-

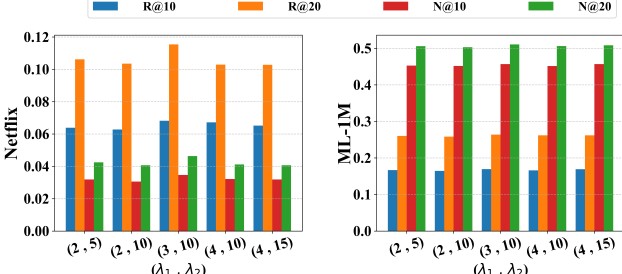

*Figure 3.* The effect of consensus weights $\lambda_1$ and $\lambda_2$

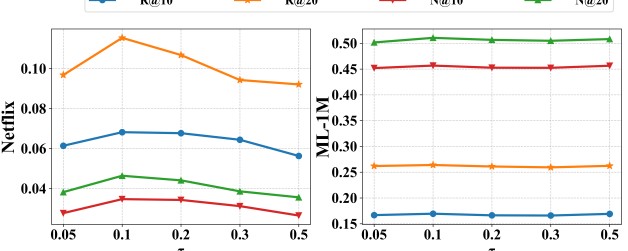

*Figure 4.* Hyperparameter experiments on the Consensus-Driven Contrastive Loss over Netflix and ML-1M datasets in terms of Recall@10, Recall@20, NDCG@10 and NDCG@20.

ent weighting settings. Experiments are conducted under various weight configurations on both the Netflix and ML-1M datasets. Results indicate that a balanced weighting between positive and negative samples yields the best performance. In particular, on Netflix, the configuration with $\lambda_1 = 3$ and $\lambda_2 = 10$ achieves a Recall@20 of 0.1154 and an NDCG@20 of 0.0464. On ML-1M, the same setting results in a Recall@20 of 0.2639 and an NDCG@20 of 0.5105, outperforming other configurations.

**Temperature parameter $\tau$ of the consensus-driven contrastive loss.** An ablation study is conducted on the temperature parameter $\tau$ of the consensus-driven contrastive loss, testing $\tau \in \{0.05, 0.1, 0.2, 0.3, 0.5\}$ on both datasets. The model performs optimally at $\tau = 0.1$, achieving Recall@20 of 0.1154 and NDCG@20 of 0.0464 on Netflix, and Recall@20 of 0.2639 and NDCG@20 of 0.5105 on ML-1M. Too small a $\tau$ overemphasizes the distinction between positive and negative samples, reducing generalization, while too large a $\tau$ weakens feature discriminability. Figure 4 illustrates how different $\tau$ values affect recommendation performance.

**Semantic threshold $\delta$.** We perform an ablation study on the preset threshold $\delta$ for semantic negative sample filtering, testing $\delta \in \{0.1, 0.2, 0.3, 0.4, 0.5\}$ on both datasets. Optimal performance is achieved at $\delta = 0.3$, with Recall@20 / NDCG@20 of 0.1154 / 0.0464 on Netflix and 0.2639 / 0.5105 on ML-1M. A small $\delta$ (0.1 or 0.2) overly restricts negative sampling, while a large $\delta$ (0.4 or 0.5) relaxes semantic discrepancy constraints, impairing sample discriminabil-

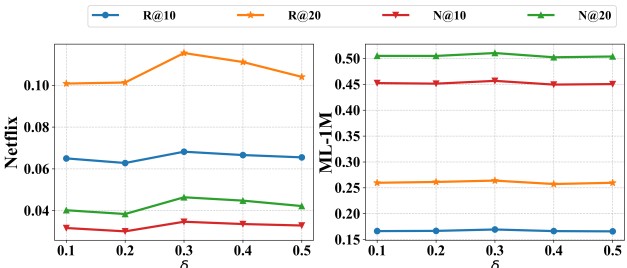

*Figure 5.* Hyperparameter experiments on the semantic negative sample threshold $\delta$ over Netflix and ML-1M datasets in terms of Recall@10, Recall@20, NDCG@10 and NDCG@20.

*Table 3.* Runtime efficiency comparison of CCLRec, CORONA, and LLMRec under the same open-source 7B LLM backbone. The last row of the Netflix table shows the detailed breakdown of CCLRec's total runtime.

| Overall Runtime on Netflix | |
| --- | --- |
| Method | Total Time (s) |
| CCLRec | 9,150 |
| LLMRec | 18,000 |
| CORONA | 12,000 |
| *CCLRec: User Emb: 3,450s, Item Emb: 300s, Train: 5,400s* | |

| Training Efficiency on ML-1M | | |
| --- | --- | --- |
| Method | Per-Epoch Time (s) | Early Stopping |
| CCLRec | 105 | Yes |
| LLMRec | 60 | Yes |
| CORONA | 100 | Yes |

ity. Figure 5 shows the performance trends for different $\delta$ values, highlighting the trade-off between negative sample validity and diversity.

### 4.5. Runtime Efficiency Analysis (RQ4)

To validate the computational efficiency of CCLRec, we compare its total runtime against CORONA and LLMRec (all utilizing the same 7B LLM backbone) on the Netflix and ML-1M datasets, as detailed in Table 3. On the Netflix dataset, CCLRec achieves a total runtime of 9,150s, reducing the time cost by approximately 23.8% compared to CORONA (12,000s) and 49.2% compared to LLMRec (18,000s). This efficiency stems from our design: although CCLRec involves dual embedding generation, the computation is predominantly focused on users, avoiding the prohibitive costs of large-scale item-side inference. While CCLRec exhibits a marginally higher per-epoch training cost on ML-1M (105s vs. 100s for CORONA and 60s for LLMRec), this slight overhead is justified by its superior convergence stability and recommendation accuracy. Overall, CCLRec achieves a superior trade-off, outperforming LLM-enhanced baselines in effectiveness while maintaining highly competitive computational efficiency.

## 5. Conclusion

In this work, we proposed CCLRec, a novel consensus-driven contrastive learning framework designed to bridge the supervisory gap between collaborative structural signals and LLM-derived semantic knowledge. Unlike existing approaches that treat structural and semantic views independently, CCLRec identifies high-confidence consensus signals endorsed by both topological proximity and semantic relevance. By integrating these signals into a weighted contrastive objective, CCLRec effectively amplifies high-confidence supervision and aligns structural proximity with semantic relevance. Extensive experiments on the Netflix and ML-1M datasets demonstrate that CCLRec consistently outperforms state-of-the-art baselines. In future work, we plan to explore more fine-grained alignment mechanisms between LLM reasoning chains and graph propagation paths, as well as investigate the scalability of CCLRec in larger-scale industrial recommendation scenarios.

## Impact Statement

This paper presents work whose goal is to advance the field of Machine Learning. There are many potential societal consequences of our work, none of which we feel must be specifically highlighted here.

## Acknowledgments

We thank all anonymous reviewers and program chairs for their constructive and helpful reviews. This work was supported by the National Natural Science Foundation of China (No. 62506371; No. 72171137), The Natural Science Foundation of Shanxi Province, China (No. 202403021222153); the Shanxi Provincial Key Laboratory (CICIP2024002); the Shanxi Higher Education Institutions Science and Technology Innovation Project (2024L164).

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

## A. LLM-based Item and User Semantic Embedding Generation

In this work, we leverage a locally deployed large language model to perform offline completion and embedding modeling of item semantic attributes, thereby constructing a stable and a reproducible semantic representation space, as shown in Figure 6. This semantic modeling process does not directly participate in the recommendation inference, it serves as the foundation for semantic correlation modeling.

### A.1. LLM-based Item Semantic Attribute Completion and Embedding Construction

In CCLRec, we design a two-stage process for item embedding generation. First, we perform semantic augmentation on the raw item attributes. For an item $v \in \mathcal{V}$, its initial attribute set is defined as:

$$T^{\text{raw}} = \{a_{v,\text{title}}, a_{v,\text{year}}\}, \tag{10}$$

where $a_{v,\text{title}}$ and $a_{v,\text{year}}$ represent specific raw features such as the title and release year. Leveraging the extensive world knowledge and reasoning capabilities of LLMs, we construct a structured prompt function $P(\cdot)$ to generate supplementary attributes. This augmentation process is formalized as:

$$T^{\text{aug}} = \text{LLM}(P(T^{\text{raw}}), \tau_{\text{llm}}), \tag{11}$$

where $\tau_{\text{llm}}$ denotes the temperature parameter, and $T^{\text{aug}}$ represents the augmented attribute set generated by the LLM. Consequently, the expanded attribute space for item $v$ is given by the union:

$$T^{\text{total}} = T^{\text{raw}} \cup T^{\text{aug}}. \tag{12}$$

Subsequently, we convert these discrete textual attributes into continuous dense embeddings using a pre-trained `Sentence-BERT` encoder, denoted as $f_{\text{enc}}(\cdot)$. For each attribute segment $k \in T^{\text{total}}$, its independent embedding is computed as $\mathbf{v}_{v,k} = f_{\text{enc}}(k)$. Finally, the unified semantic representation of item $v$ is obtained by aggregating these attribute embeddings:

$$e_v^{\text{llm}} = \text{Agg}(\mathbf{v}_{v,k} \mid k \in T^{\text{total}}), \tag{13}$$

where $\text{Agg}(\cdot)$ denotes the feature aggregation function (e.g., mean pooling).

$$e_v = e_v^{\text{base}} + w_{\text{llm}} e_v^{\text{llm}} + w_{\text{CLIP}} e_v^{\text{CLIP}}. \tag{14}$$

The weights $w_{\text{llm}}$ and $w_{\text{clip}}$ are set following the baseline method LLMREC. $e_v^{\text{base}}$ is the ID embedding. $e_v^{\text{CLIP}}$ is the item image embedding. We process poster images in batches using a local CLIP vision encoder to extract dense visual embeddings. The resulting embeddings are normalized and

stored in a feature matrix with shape (max ItemID $+ 1, d$), where $d$ denotes the CLIP embedding dimension. These embeddings are subsequently used as visual features for downstream multimodal recommendation models.

### A.2. LLM-based User Semantic Embedding Generation

Relying solely on the user-item interaction structure makes it challenging to capture users' long-term preferences at the semantic level. To address this, in addition to structural modeling, we introduce a mechanism to construct user semantic representations based on item semantic attributes. Leveraging the contextual reasoning capabilities of LLMs, historical user interactions are transformed into structured natural language descriptions and further mapped into continuous semantic embeddings.

Specifically, the user's semantic representation is generated based on their historical interactions. For a given user $u$, let the historical interaction set be:

$$\mathcal{H}_u = \{v_1, v_2, \ldots, v_q\}, \tag{15}$$

where $v_i$ is item. We map the user's most recent segment of historical interactions into a sequence of textual attributes including year, title, and category, and construct a prompt function to generate the user semantic text $R_u$:

$$W_u = \text{LLM}(\mathcal{H}_u; \tau_{\text{llm}}), \tag{16}$$

The generated text object $R_u$ is then flattened into a continuous natural language description $D_u$. Subsequently, a `Sentence-BERT` encoder is applied to extract the initial semantic embedding of the user:

$$e_u = f_{\text{enc}}(D_u). \tag{17}$$

The above procedure yields semantically enriched user and item representations, which serve as the foundation for subsequent recommendation modeling.

## B. Dataset Statistics

To ensure a fair comparison, we directly use the preprocessed and released multimodal Netflix dataset from Wei et al. (2024), which augments items in the Netflix Prize data with crawled movie posters and rich metadata (e.g., titles, release years, and genres). For the widely used MovieLens-1M (ML-1M) benchmark (Cui et al., 2025), which does not provide native visual information, we adopt the same multimodal data construction pipeline to acquire the corresponding movie posters, ensuring consistency across datasets. The statistics of both datasets are summarized in Table 4.

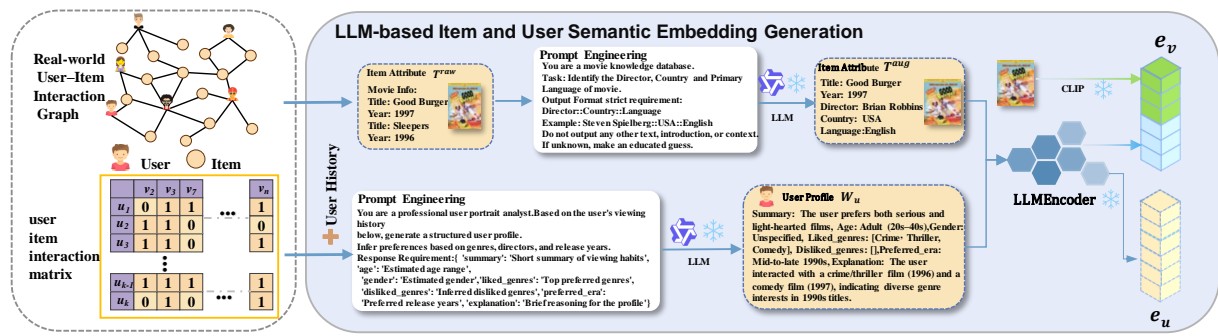

*Figure 6.* Illustration of the LLM-based semantic embedding generation process.

*Table 4.* Statistics of the evaluated datasets.

| Statistic | Netflix | ML-1M |
|---|---|---|
| #Users | 13,187 | 6,040 |
| #Items | 17,366 | 3,260 |
| #Interactions | 70,778 | 998,539 |
| Density | 0.0308% | 5.06% |
| Avg. Int./User | 5.37 | 165.33 |
| Avg. Int./Item | 4.08 | 306.31 |

## C. Visual Data Preparation

Poster images for all items are collected from The Movie Database (TMDB) via its public API.[1] For each item, search queries are constructed based on the item title after basic normalization (e.g., removing parentheses and redundant phrases). To maximize coverage, a multi-strategy retrieval scheme is applied: if exact matches fail, progressively simplified title variants are used. Among retrieved candidates, the first result with an available poster is selected and downloaded. For items whose posters cannot be retrieved from TMDB, a uniform placeholder image is used, consisting of a black background with the item title displayed in white text at the center, ensuring consistent visual input. The statistics of poster acquisition for the evaluated datasets are summarized in Table 5.

*Table 5.* Poster Acquisition Statistics for Different Datasets

| Statistic | ML-1M | Netflix |
|---|---|---|
| Total Items | 3260 | 17366 |
| Successfully Downloaded | 2914 | 14826 |
| Missing Posters | 346 | 2540 |
| Proportion (%) | 89.3 / 10.7 | 85.4 / 14.6 |

## D. Multimodal Feature Integration in User–Item Graph Embeddings

The extracted image features, user embeddings, and item attribute embeddings are projected into a unified embed-

[1] https://www.themoviedb.org/

ding space and propagated through the user–item interaction graph. During forward propagation, image features are fused with user and item embeddings via graph propagation, while item textual attribute embeddings are also incorporated. This process produces comprehensive user and item representations that integrate graph structure information, multi-modal features, and item attributes for downstream recommendation tasks.

## E. Hyperparameter Sensitivity

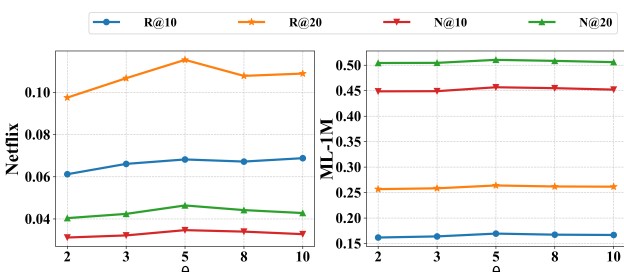

*Figure 7.* Hyperparameter experiments on the structural co-occurrence threshold $\theta$ over Netflix and ML-1M datasets in terms of Recall@10, Recall@20, NDCG@10 and NDCG@20.

**Structural co-occurrence threshold $\theta$.** We conduct an ablation study on the structural co-occurrence threshold $\theta$, testing $\theta \in \{2, 3, 5, 8, 10\}$ on both the Netflix and ML-1M datasets. The results show that the model achieves optimal performance at $\theta = 5$, with Recall@20 / NDCG@20 of 0.1154 / 0.0464 on Netflix and 0.2639 / 0.5105 on ML-1M. Smaller $\theta$ values (e.g., 2 or 3) result in weaker discriminative structural negative samples, while larger values (e.g., 8 or 10) reduce the number of effective negative samples. Figure 7 shows the performance trends for different $\theta$ values, illustrating the impact of the threshold on recommendation performance.

