# OpenReview forum: "CCLRec: Consensus-driven Contrastive Learning for LLM-enhanced Graph Recommendation"
_ICML.cc/2026/Conference — ICML 2026 regular_

### Official Review · Reviewer_Wh6h · 2026-02-22

**Soundness:** 2
**Presentation:** 3
**Significance:** 2
**Originality:** 2
**Overall Recommendation:** 4
**Confidence:** 4

**Summary:**

This paper proposes CCLRec, a consensus-driven contrastive learning framework for LLM-enhanced graph recommendation. The key idea is to explicitly mine structural–semantic consensus between graph-based collaborative signals and LLM-derived semantic similarities, and to use these consensus samples as high-confidence supervision in a weighted contrastive learning objective.

**Compliance With Llm Reviewing Policy:**

Affirmed.

**Key Questions For Authors:**

Please refer to Strengths And Weaknesses part.

**Limitations:**

Please refer to Strengths And Weaknesses part.

**Strengths And Weaknesses:**

1. My main concern will be technical improvement. Based on my understanding, all of involved techniques are old or existing one, just combine them to improve performance. Is there any technical improvement that author can highlight?
2. The approach relies on multiple thresholds and hyperparameters (e.g., δ, θ, s, λ₁, λ₂). Although sensitivity studies are provided, the method may require careful tuning in practice, which could limit robustness across domains.
3. The quality of semantic sampling depends heavily on the correctness and stability of LLM-generated embeddings and attribute completion. Potential biases or hallucinations could be introduced by the LLM.
4. The semantic representations are generated using LLMs with substantial prior knowledge of the movie domain (e.g., genres, directors, eras). This raises the question of whether part of the observed performance gain comes from domain memorization rather than genuine alignment between collaborative and semantic signals.

---

> ### Author Rebuttal · Authors · 2026-03-30
>
> **W1:**
>
> We appreciate this constructive suggestion. While our framework builds on established components (GNNs, LLMs, CL), its core technical contribution is a mutual denoising and consensus mechanism that resolves the supervisory conflict between topological proximity and semantic relevance. Specifically, we highlight the following improvements:
>
> - **Our structure-semantic consensus mining strategy computes the strict intersection of structural and semantic candidate sets.**  This mutual error-correction explicitly filters out structural spurious edges and LLM hallucinations, ensuring the construction of high-confidence positive and negative samples for our contrastive objective.
>
> - **Unlike conventional InfoNCE formulations, our consensus-driven contrastive objective is specifically designed to focus the learning process on high-confidence consensus pairs.** By introducing weighting ($\lambda_1, \lambda_2$) to assign higher importance to reliable consensus positives and strongly penalize consensus negatives (Eq. 6-7), the tailored loss successfully resolves the inability of standard methods to distinguish the relative significance of different multi-view pairs.
>
> - **Beyond the scope of recommendation, the bidirectional mutual-denoising mechanism of CCLRec serves as a methodological inspiration for reconciling conflicting structural and semantic signals.** Furthermore, it provides actionable insights for grounding LLM generation and mitigating hallucinations via deterministic graph topologies.
>
> **W2:**
>
> Regarding the concern about hyperparameter sensitivity, empirical evidence demonstrates that our method is highly robust and does not require exhaustive dataset-specific tuning:
>
> - **Our core mechanism is inherently robust across datasets with drastically different topological densities, with the sampling size consistently set to $s=2$ and the contrastive weights ($\lambda_1, \lambda_2$) demonstrating high stability.** This consistent performance on both the highly sparse Netflix and the dense ML-1M datasets proves that exhaustive dataset-specific tuning is unnecessary.
>
> - **The hyperparameters ($\delta$ and $\theta$) follow clear, predictable trade-offs, exhibiting consistent inverted-U performance curves across different datasets.** As demonstrated in our parameter analysis, setting these thresholds too low (e.g., $\theta \le 3$, $\delta \le 0.2$) applies overly strict criteria, yielding "easy" negative samples that provide weak optimization gradients. Conversely, setting them too high (e.g., $\theta \ge 8$, $\delta \ge 0.4$) excessively relaxes the boundaries, inadvertently introducing false negatives into the consensus set. **Therefore, moderate values ($\theta=5$, $\delta=0.3$) universally achieve the optimal balance between bounding the negative sampling space and maintaining strong sample discriminability.**
>
> **W3:**
>
> As highlighted in our manuscript, while CCLRec is primarily designed to resolve the fundamental conflict between structural and semantic information, this consensus-driven architecture also naturally mitigates the risks of LLM hallucinations and inherent biases.
>
>  Rather than relying solely on LLM-generated semantics, we utilize the real-world interaction graph as a reliable empirical reference. Through our structure-semantic consensus mining strategy (i.e., $\mathcal{P}_c^u = \mathcal{P}_s^u \cap \mathcal{P}_t^u$), a candidate is included in the high-confidence consensus set when it is supported by both LLM semantic reasoning and topological proximity. Under this dual-validation mechanism, hallucinated or biased samples generated by the LLM typically lack structural support. **Consequently, our consensus mechanism naturally filters out these unverified outliers, reducing the interference of unreliable LLM outputs on the model's representation space.** We will clarify this mechanism in the revised Methodology section.
>
> **W4:**
>
> Regarding the concern about domain memorization, empirical evidence confirms that CCLRec's performance gains stem from our consensus-driven alignment rather than mere reliance on LLM prior knowledge:
>
> - **Since all LLM-augmented baselines utilize identical semantic features, CCLRec's superior performance clearly stems from our consensus mechanism, not domain memorization.** To ensure this fair comparison, we directly used the public benchmark data for Netflix and uniformly generated representations via a local Qwen-7B model for ML-1M.
>
> - If performance gains were driven by domain memorization, relying solely on semantic signals would yield comparable results. However, our ablation study proves otherwise. **As demonstrated in Table 2, removing the structural view (w/o structure) or the intersection mechanism (w/o consensus) significantly degrades performance.** This explicitly confirms that the mutual validation between topological signals and semantic reasoning is the true source of our improvements.

---

> > ### Author Rebuttal · Reviewer_Wh6h · 2026-04-01
> >
> > Thanks for your response authors. After checking other reviewers' score and the rebuttal provided, I am happy to raise my score to weak accept. Overall, I will say the idea is interesting but technical side needs more improvements. Since it is a conference, the main thing is to share the new/interesting idea, so I will raise my score, but if authors want to submit to a journal as extension version of this paper, please add more technical improvement part.

---

> > > ### Author Response · Authors · 2026-04-02
> > >
> > > We sincerely thank the reviewer for the positive feedback and for increasing the score to Weak Accept. We are glad that the reviewer found our core idea interesting and suitable for sharing within the conference community.
> > >
> > > Regarding the suggestion for future work, we completely agree that further technical refinements would strengthen the paper for a journal extension. We will take your constructive comments into account and dedicate more effort to enhancing the technical depth in the journal extended version. We believe these improvements will provide a more comprehensive study of the proposed method.
> > >
> > > Thank you again for your valuable time and insightful suggestions throughout the review process.

---

### Official Review · Reviewer_o1Zm · 2026-03-09

**Soundness:** 3
**Presentation:** 4
**Significance:** 4
**Originality:** 3
**Overall Recommendation:** 5
**Confidence:** 4

**Summary:**

This paper studies a natural problem in LLM-enhanced recommendation: structural signals from interaction graphs and semantic signals derived from textual/LLM-based information are not always consistent, which may introduce conflicting supervision during representation learning. To address this, the authors propose a contrastive learning framework based on consensus samples. The main idea is to construct neighbors from structural and semantic views separately, and then use the samples supported by both views as higher-confidence supervision signals. Experiments on two benchmark datasets show consistent improvements over several baselines, and the ablation and sensitivity results provide initial support for the contribution of the main components.

**Compliance With Llm Reviewing Policy:**

Affirmed.

**Key Questions For Authors:**

1. How does the method behave when the overlap between structural neighbors and semantic neighbors is small? For example, have the authors observed any particular behavior for users with short or sparse interaction histories?
2. The paper currently uses a hard-intersection strategy to construct consensus samples. Could the authors further explain why this choice is more appropriate than union-based or softer agreement-based alternatives, or what practical advantages it offers?
3. There are also a few typos and minor writing issues. For example, CKE on Page 2 is mentioned without a citation.

**Strengths And Weaknesses:**

Strengths
1. The paper addresses a problem with clear practical relevance.
It focuses on a natural yet important issue in LLM-enhanced recommendation, namely that collaborative structural signals and semantic signals may not always be consistent and can even conflict with each other. This problem setting is well motivated and gives the proposed method a meaningful foundation.
2. The overall method is clearly structured and intuitively designed.
The framework first constructs samples from the structural and semantic views, and then uses a consensus mechanism to identify high-confidence supervision signals. This overall pipeline is easy to follow, and the idea of using consensus samples to alleviate multi-view supervision conflicts is conceptually simple and interpretable.
3. The experimental results are reasonably comprehensive and provide initial support for the method’s effectiveness.
The paper evaluates the proposed approach against multiple baselines on two widely used datasets, and also includes ablation studies and hyperparameter sensitivity analysis. Although the mechanism-level evidence could be further strengthened, the current results already provide a reasonable empirical basis for the practical effectiveness of the method.

Weaknesses
1. The core idea of the paper is that consensus samples can provide higher-quality supervision, and I think this intuition is reasonable. However, the current experiments mainly show that the method is effective, while the direct evidence for why it works is still somewhat limited. The paper would be stronger if it could include more direct analysis beyond final performance, for example by comparing the quality of consensus samples with that of single-view samples.
2. Using the intersection to define consensus samples is intuitive and easy to understand. That said, it would still be helpful for the paper to further explain why this relatively strict hard-intersection strategy is adopted, rather than a softer agreement-based alternative.
3. The motivation that structural signals and semantic signals may conflict is convincing, but at the moment this part remains somewhat intuitive. It would strengthen the paper if the authors could provide more analysis on when the two views disagree the most, or under what scenarios the proposed method tends to bring larger gains.

---

> ### Author Rebuttal · Authors · 2026-03-30
>
> **W1:**
>
> We appreciate this suggestion. Our ablation study (Table 2) clearly shows the high quality of consensus samples through three observations:
>
> - **CCLRec significantly outperforms the basic multi-view model that uses all samples without filtering (w/o consensus).** This shows that our strategy successfully removes noise and extracts better training signals.
>
> - **CCLRec consistently surpasses both single-view baselines (w/o semantic and w/o structure).** This means consensus samples effectively combine the strengths of both views, providing more reliable supervision.
>
> - **Our results provide clear evidence of view conflict on the sparse Netflix dataset.** Specifically, directly combining both views (w/o consensus) actually drops performance (e.g., Recall@10 drops to 0.0639) compared to using just one view (e.g., w/o semantic achieves 0.0655). This drop proves that unfiltered multi-view samples contain conflicting noise. CCLRec fixes this issue and boosts performance, proving that our mechanism successfully isolates high-quality, mutually verified samples.
>
> We will add this analysis to the revised manuscript to explain why consensus samples work better.
>
> **W2Q2:**
>
> We sincerely thank the reviewer for this insightful question. We adopted the strict hard-intersection strategy primarily to guarantee absolute high-confidence mutual denoising.
>
> A union-based or softer agreement strategy would inevitably retain unverified signals, such as spurious graph edges or hallucinated LLM semantics, thereby reintroducing the exact supervisory conflict our framework aims to eliminate. By enforcing a strict hard intersection, we guarantee that every consensus sample is dually validated by both topological proximity and semantic reasoning. **This strict boundary acts as a highly reliable reference, providing the unambiguous and high-quality supervisory signals essential for effective contrastive learning.**
>
>
> **W3:**
>
> To clarify when structural and semantic signals conflict and where our method excels, we analyzed two key scenarios:
>
> - **Sparsity inherently exacerbates view conflict, as demonstrated by our gains on the highly sparse Netflix dataset.** Sparse graphs often contain noisy edges from random clicks, while LLMs, lacking historical context, tend to generate biased semantic features based on prior knowledge. CCLRec’s dual-validation mechanism effectively resolves this pronounced divergence by filtering out both unsupported structural noise and semantic deviations.
>
> - When **user interests diverge**, relying solely on LLM semantics often introduces false positives，items that are textually similar but rarely co-occur in actual collaborative filtering. In such cases, the structural view provides a critical constraint. Our consensus mechanism successfully resolves this by isolating semantic candidates that are explicitly backed by genuine topological evidence.
>
> We will incorporate this analysis of applicable scenarios into the revised manuscript.
>
>
> **Q1:**
>
> We appreciate this insightful question. We have analyzed the behavior of small overlaps from two perspectives:
>
> - **When the overlap is empty (e.g., $\mathcal{P}_c^u = \emptyset$), the model uses a fallback mechanism, temporarily returning to standard dual-view training to maintain stability.** To analyze this, we recorded the probability of empty positive and negative consensus sets within training batches of 1024 samples. As shown in the table below, a setting like $s=1$ frequently causes empty sets (e.g., a 54% empty rate for positive sets on Netflix). This forces the model to rely too much on the fallback, reducing overall performance. Thus, an appropriate sampling size ($s \ge 2$) is needed to ensure sufficient overlap.
> | s | 1 | | 2 | | 3 | | 4 | | 5 | |
> | --- | --- | --- | --- | --- | --- | --- | --- | --- | --- | --- |
> | NETFLIX | 54% | 76% | 31% | 73% | 3% | 70% | 1% | 69% | 1% | 66% |
> | ML-1M | 80% | 82% | 75% | 78% | 60% | 75% | 56% | 77% | 52% | 75% |
>
> - **For users with sparse histories, graph noise and LLM semantic errors naturally cause a small overlap.** However, this actually shows that our strict intersection successfully filters out conflicting noise, keeping only highly reliable signals. Because of this effective noise filtering, CCLRec achieves its most significant relative gains on highly sparse datasets (like Netflix), solving the view conflict for long-tail users.
>
> **Q3:**
>
> We sincerely thank the reviewer for their meticulous review. We will add the correct citation for CKE on Page 2 in the final version. Furthermore, we will conduct a thorough proofreading of the entire manuscript to correct all typos and minor writing issues, thereby further improving the overall quality of the paper.

---

> > ### Author Rebuttal · Reviewer_o1Zm · 2026-04-04
> >
> > Thanks for the response and I will keep my positive score.

---

### Official Review · Reviewer_KuxA · 2026-03-12

**Soundness:** 1
**Presentation:** 2
**Significance:** 3
**Originality:** 3
**Overall Recommendation:** 2
**Confidence:** 3

**Summary:**

This paper proposes CCLRec, a consensus-driven contrastive learning framework that integrates graph-structural signals and LLM-derived semantic information for recommendation. The method leverages consensus positives and negatives by intersecting semantic and structural neighborhoods. It uses a weighted contrastive objective alongside a standard BPR loss to emphasize high-confidence signals. Experiments on Netflix and ML-1M show consistent gains over a range of CF, multimodal, and recent LLM-based baselines, with ablations indicating contributions from each component.

**Compliance With Llm Reviewing Policy:**

Affirmed.

**Key Questions For Authors:**

Q1: regarding the defined loss, is there a single anchor per positive pair, or are you contrasting item–item pairs against pairwise negatives? How are in batch negatives incorporated?

Q2: can you specify the adopted evaluation protocol? Please detail train/test/val split strategy, how do you avoid leakage when LLM processes user histories, the amount of test positives per user, etc.

Q3: can you release the code? If not, please provide configuration details (random seeds, epoch counts, early stopping criteria, etc.) to reproduce your results.

**Limitations:**

yes

**Strengths And Weaknesses:**

Strenghts:

- The main idea of cross-view consensus sets is intuitive and well motivated.
- The pipeline is clearly explained: sampling in both views, consensus mining, and training objectives.
- The work addresses a critical issue in LLM-enhanced graph recommendation by aligning structural and semantic signals to minimize conflicts.

Weaknesses:

-The contrastive loss formulation is ambiguous and potentially unsound: Eq. (6)–(7) operate over item–item pairs without a clear anchor, yet the text refers to “anchor user u”; normalization over pairwise negatives is unusual and needs clearer justification or correction.
- Fairness: there is no training for baselines on Netflix dataset. Results in Table 2 are taken from original papers.
- Clarity: many references are missing during the presentation (e.g. in introduction)
- Reproducibility: the absence of runnable code is one of the major drawback. Moreover, details on evaluation protocol (e.g. data split) are missing. NDCG values on MovieLens are unusually high.

---

> ### Author Rebuttal · Authors · 2026-03-30
>
> **W1Q1:**
>
> We sincerely appreciate the reviewer for highlighting this notational ambiguity. A disconnect existed between the phrase "anchor user u" and the Eq. (6)-(7), leading to confusion.  **The contrastive operation is performed on item-item pairs within the localized context of a specific user $u$. To mathematically clarify this, the consensus sets are strictly redefined with a user-specific superscript (i.e., $\mathcal{P}_c^u$ and $\mathcal{N}_c^u$).**
>
> Specifically, for each user $u$, $\mathcal{H}_u$ denotes the sequence of items in their historical interactions. We construct the user-specific consensus positive and negative sets based on $\mathcal{H}_u$. $u$ serves as the index variable to determine the positive and negative item spaces, without participating in the dot-product optimization as an anchor. Instead, the contrastive objective treats every item $v$ as an anchor, and its positive counterparts are the other consensus items $v' \in \mathcal{P}_c^u \setminus \{v\}$ preferred by the same user. Thus, we pull the representations of items mutually endorsed by both structural and semantic views closer together, creating a more compact item-side representation space. The consensus-driven contrastive loss is corrected as follows:
>
> $l\_{CCL}^{u} = -\lambda\_1 \sum\_{v,v' \in \mathcal{P}\_c^u} \log \frac{\exp(e\_v^\top e\_{v'} / \tau)}{Z\_u(v)} - \sum\_{v,v' \in (\mathcal{P}\_t^u \cup \mathcal{P}\_s^u) \setminus \mathcal{P}\_c^u} \log \frac{\exp(e\_v^\top e\_{v'} / \tau)}{Z\_u(v)}$
>
> Where the normalization term $Z_u(v)$ is formulated as follows:
>
> $Z\_u(v) = \lambda\_2 \sum\_{v^- \in \mathcal{N}\_c^u} \exp(e\_v^\top e\_{v^-} / \tau) + \sum\_{v^- \in (\mathcal{N}\_t^u \cup \mathcal{N}\_s^u) \setminus \mathcal{N}\_c^u} \exp(e\_v^\top e\_{v^-} / \tau)$
>
> **W2:**
>
> Regarding baseline comparisons on the Netflix dataset: To provide a fairer comparison, we re-ran the baselines locally based on your suggestion. Notably, despite our best efforts, our local reproduction of CORONA yielded exceptionally low results (0.0189, 0.0417, 0.0082, 0.01387). To ensure a strictly fair evaluation and avoid penalizing the strongest baseline, we report CORONA's official published results. Our local reproduction results for the remaining methods are as follows:
>
> | Method | R@10 | R@20 | N@10 | N@20 |
> | :--- | :--- | :--- | :--- | :--- |
> | MF-BPR | 0.0336 | 0.0536 | 0.0147 | 0.0200 |
> | NGCF | 0.0292 | 0.0460 | 0.0146 | 0.0189 |
> | LIGHTGCN | 0.0417 | 0.0710 | 0.0187 | 0.0261 |
> | NFM | 0.0246 | 0.0368 | 0.0239 | 0.0301 |
> | CKE | 0.0420 | 0.0677 | 0.0297 | 0.0426 |
> | LATTICE | 0.0271 | 0.0742 | 0.0119 | 0.0161 |
> | MICRO | 0.0466 | 0.0758 | 0.0226 | 0.0301 |
> | MMSSL | 0.0422 | 0.0715 | 0.0186 | 0.0249 |
> | LLMREC | 0.0509 | 0.0807 | 0.0258 | 0.0333 |
> | CORONA | 0.0616 | 0.0938 | 0.0279 | 0.0416 |
> | CCLREC (OURS) | **0.0682** | **0.1154** | **0.0347** | **0.0464** |
>
> **As shown, our locally re-run results are largely consistent with the previously recorded experimental results. We will update Table 1 with these results in the final manuscript.**
>
> **W3:**
>
> We will update the Introduction with the missing citations to substantiate our contextual claims:
>
> (1) **On graph propagation failures under sparsity and noise:**
>
> - Are graph augmentations necessary? Simple graph contrastive learning for recommendation (SIGIR, 2022)
>
> - Mitigating Extreme Cold Start in Graph-based RecSys through Re-ranking（CIKM, 2024）
>
> - Distributionally robust graph-based recommendation system (WWW, 2024)
>
> (2) **On LLMs and semantics in recommendation:**
>
> - Semantics-guided disentangled learning for recommendation (TKDE, 2022)
>
> - Large language models meet causal inference: Semantic-rich dual propensity score for sequential recommendation (TKDE, 2025)
>
> - Bridging the User-side Knowledge Gap in Knowledge-aware Recommendations with Large Language Models (AAAI, 2025)
>
> **Q2:**
> - We evaluate performance using Recall@K and NDCG@K. Recall@K measures the proportion of true interactions retrieved, while NDCG@K evaluates ranking quality by emphasizing top positions.
> - To ensure a fair evaluation protocol, we directly adopted the **public benchmark** (LLMRec) for the Netflix dataset. **For the ML-1M dataset, we followed the 8:2 splitting strategy of CoLaKG (80% for training and 20% for testing).**
> - **To strictly prevent data leakage, test set interactions were completely masked when constructing prompts and generating semantic representations.** The LLM was exclusively exposed to the training set, which fundamentally eliminates this risk.
>
> **W4Q3:**
>
> - **Complete source code, datasets, and configurations were provided in our initial supplementary materials, enabling direct reproduction of all results.**
> - **The high NDCG values on the ML-1M dataset are not anomalous.** The baselines also achieve highly competitive results on this benchmark (e.g., CORONA at 0.5027 and LLMRec at 0.4970). Our performance (0.5105) represents a reasonable increment over these methods.

---

> > ### Author Rebuttal · Reviewer_KuxA · 2026-04-03
> >
> > I thank the authors for their rebuttal. However, after carefully reviewing the authors' responses and conducting a deep inspection of the provided supplementary source code, I find that my core concerns regarding reproducibility, soundness, and fairness of comparison **have not been resolved**. In fact, inspecting the code has revealed severe discrepancies between the paper's claims and the actual implementation.
> >
> > # Reproducibility #
> > I am very sorry for not noticing that the code was in the supplementary material. Anyway, the submitted code is fundamentally broken and **cannot be directly executed**. Inspecting the repository, the code contains immediate, fatal crashes. After spending some time, I was able to fix it and run it.
> >
> > # Visual modality #
> > The paper’s title, abstract, and methodology (Section 3) heavily market the framework as an "LLM-enhanced" recommendation system, focusing entirely on textual semantics and graph structure.
> > However, the source code reveals that this is actually a Multimodal GNN. The code explicitly loads image_feats.npy (CLIP embeddings of movie posters), propagates them through specific image-user graphs, and directly adds them to the final user and item embeddings (u_g_embeddings = u_g_embeddings + ... image_user_feats). **The visual modality is relegated to the Appendix in the paper and omitted entirely from the main methodological claims**.
> > This introduces a fatal flaw in the evaluation. By utilizing high-quality visual features, CCLRec has a **massive data advantage over standard CF baselines** (LightGCN, BPR) and **text-only LLM baselines** (LLMRec, CORONA), which do not have access to these images. **The ablation study (Table 2) does not ablate the visual modality**. Therefore, it is **impossible to tell if the performance gains come from the proposed "consensus-driven LLM mechanism" or simply from the fact that the model is secretly leveraging visual movie posters.**
> >
> > # Math and code #
> >
> > The authors kindly corrected the loss formulation in the rebuttal to clarify the anchor mechanism. However, the implementation does not reflect a principled mathematical derivation.
> > In the code, the "consensus weighting" is implemented as a **hardcoded heuristic step-function deep inside the training loop **(e.g., INTERSECT_W = 3 and INTERSECT_W_1 = 10.0), applied blindly to the InfoNCE logits if a boolean intersection is found.
> > More importantly, the authors claim this consensus mechanism solves "prevalent interaction sparsity." Yet, mathematically, if a user is a cold-start user, their structural neighborhood will be sparse or empty. Based on Eq. 5, the consensus intersection will also be empty. **As the authors admit, the weight then vanishes. Therefore, the core novelty of this paper logically deactivates in the exact high-sparsity scenarios it claims to solve.**
> >
> > I am maintaining my recommendation to **reject**.

---

> > > ### Author Response · Authors · 2026-04-04
> > >
> > > **For Visual modality**
> > >
> > > Thank you very much for the detailed feedback. We think the concerns about baseline fairness and visual feature integration come more from a lack of clarity in our writing than from any actual flaw in the method.
> > >
> > > - **For the Netflix dataset, we strictly used the public version released by LLMRec.** More importantly, we re-ran and evaluated all baseline methods on this exact same dataset.
> > >
> > > - **We applied the same data construction pipeline to ML-1M, and evaluated all baselines using this dataset.** Since every model received exactly the same input features, the performance gains from CCLRec **only come from** our consensus-driven mechanism, **not** from any asymmetric data advantage.
> > >
> > > - Crucially, our design actually follows the same paradigm established by LLMRec, which already combines visual and textual features into a holistic item embedding. In our framework, **visual signals are treated as an intrinsic part of the item's semantic information.** We acknowledge this design point was not clearly explained in the original manuscript. Following LLMRec, we will revise the manuscript to explicitly state that our semantic representations include visual features.
> > >
> > > **For Math and code**
> > >
> > > Thank you very much for your valuable suggestions.
> > >
> > > - First, we would like to respectfully clarify the **central focus** of our paper. Our **core contribution** lies not in solving the “prevalent interaction sparsity” problem, but rather in bridging the supervisory gap between structural and semantic signals. Furthermore, our mechanism is capable of handling relatively sparse data effectively for recommendation tasks. We will actively revise any wording in the manuscript that may have caused misunderstanding to eliminate ambiguity.
> > >
> > > - Our empirical results clearly refute the assumption that our mechanism would fail under sparsity. As shown in the dataset statistics (Table 4), the Netflix dataset is sparse, with an average of **only 5.37 interactions per user.** Given the training split, the vast majority of users have only 4 to 6 interactions available during training. Even with such limited data, CCLRec still achieves significant performance improvements (e.g., improving Recall@20 from CORONA’s 0.0938 to 0.1154).
> > >
> > > - To further validate this point, we conducted an additional experiment on the Netflix dataset: **we retained only items that appear at most twice in the global graph as our training set.** The results show that the consensus mechanism still brings performance gains. The reason is that, despite sparse interactions, as long as consensus is reached on both the structural and semantic sides, the model can amplify these high‑confidence signals, thereby effectively avoiding the noise‑dominated collapse that occurs in sparse learning.
> > >
> > > |         | R@10   | R@20   | N@10   | N@20   |
> > > |---------|--------|--------|--------|--------|
> > > | LLMREC  | 0.0119 | 0.0296 | 0.0073 | 0.0142 |
> > > | CCLREC  | 0.0187 | 0.0379 | 0.0091 | 0.0182 |
> > >
> > > - We believe that the core idea of CCLRec can be naturally extended to handle the cold‑start problem in future work. When graph structure edges are **completely missing**, we can adapt the framework by leveraging multimodal information. For example, by aligning the visual space and the textual semantic space of items, we can construct cross‑modal consensus positive and negative samples. This demonstrates the extensibility of our consensus-driven design. We will include a discussion of this potential application in the revised manuscript.
> > >
> > > **We once again thank the reviewer for the constructive feedback, which greatly helps us refine our manuscript and present our contributions more clearly.**

---

### Official Review · Reviewer_16Pe · 2026-03-12

**Soundness:** 4
**Presentation:** 3
**Significance:** 4
**Originality:** 3
**Overall Recommendation:** 5
**Confidence:** 5

**Summary:**

This paper proposes CCLRec, a consensus-driven contrastive learning framework designed to resolve the supervisory conflict between structural collaborative signals and LLM-derived semantic knowledge in graph-based recommendation systems. To bridge this gap, the authors introduce a structure-semantic collaborative consensus mining strategy that identifies high-confidence positive and negative sample pairs by computing the intersection of a node's structural neighbors and its semantically similar items. By integrating these consensus pairs into a weight-aware contrastive loss objective, the model adaptively amplifies the contribution of high-quality, cross-view aligned features during training. Extensive experiments on the Netflix and ML-1M benchmark datasets demonstrate that CCLRec effectively mitigates multi-view noise and significantly outperforms state-of-the-art baselines

**Compliance With Llm Reviewing Policy:**

Affirmed.

**Final Justification:**

My concerns have been addressed, and I have raised my rating.

**Key Questions For Authors:**

1.	The method for handling empty consensus sets ($\mathcal{P}_c$) is mentioned but the impact of this fallback mechanism on overall performance is not deeply analyzed. How often does this occur, and how much does it affect results?
2.	Some terminology shifts between "structure-semantic" and "structural-semantic" inconsistently throughout the paper.
3.	The structural negative sampling threshold ($\theta$) selection criterion is described but not thoroughly justified, why $\theta=5$ is optimal could use more explanation.

**Limitations:**

The authors have included an Impact Statement, but the main text and appendix currently lack a dedicated Limitations section. The authors should explicitly discuss the limitations of their work. For example, they could address the computational overhead required for the offline LLM semantic embedding generation .

**Strengths And Weaknesses:**

# Strengths
1.	The paper proposes a technically sound framework with clear problem motivation, addressing the conflict between structural and semantic information in LLM-enhanced graph recommendation. The proposed consensus-driven sampling strategy is logically coherent and directly tackles the stated challenge.
2.	The paper is well-structured with a clear logical flow.
3.	Experimental results show substantial improvements over strong baselines, demonstrating practical utility.

# Weaknesses
1.	Some sentences are overly long and complex, making them difficult to follow.
2.	The paper lacks a clear limitations section or discussion of failure cases, which would strengthen the presentation by showing balanced assessment.

---

> ### Author Rebuttal · Authors · 2026-03-30
>
> **W1:**
>
> We will comprehensively revise the manuscript, splitting overly complex sentences to improve the paper's overall readability.
>
> **W2:**
>
> One limitation of our method is that **computational efficiency decreases for users with excessively long interaction histories**, as the overhead for calculating structural co-occurrences and semantic similarities increases with sequence length. In practice, we mitigate this scalability issue by truncating the input to the user's most recent interactions, which we will explicitly discuss in the revised Limitations section.
>
> **Q1:**
>
> We sincerely thank the reviewer for this question. **To analyze this, we recorded the probability of empty consensus sets within training batches of 1024 samples.**
>
> | s | 1 | | 2 | | 3 | | 4 | | 5 | |
> | :--- | :--- | :--- | :--- | :--- | :--- | :--- | :--- | :--- | :--- | :--- |
> | NETFLIX | 54% | 76% | 31% | 73% | 3% | 70% | 1% | 69% | 1% | 66% |
> | ML-1M | 80% | 82% | 75% | 78% | 60% | 75% | 56% | 77% | 52% | 75% |
>
> As shown in our statistical table, an overly strict sampling size like $s=1$ results in a high probability of empty sets (e.g., reaching 54% and 76% for positive and negative sets on Netflix). **This severe lack of valid consensus samples directly deprives the model of essential supervisory signals, causing suboptimal performance**. As $s$ increases to 2, the empty rate drops, providing sufficient high-quality consensus pairs. However, setting $s$ too large artificially inflates the consensus set and introduces unverified noise, which also degrades performance. Therefore, $s=2$ achieves the optimal balance.
>
> **Q2:**
>
> We sincerely thank the reviewer for pointing out this oversight. We will perform a global search and replace in the revised manuscript to strictly unify all related terminology to 'structure-semantic,' ensuring academic consistency.
>
> **Q3:**
>
> We will thoroughly justify the selection of $\theta=5$ in the revised manuscript by clarifying the underlying trade-off in negative sample construction. Specifically, the threshold $\theta$ controls the boundary of structural irrelevance. If $\theta$ is too small (e.g., 2 or 3), the selection criterion becomes overly strict, yielding "easy" negative samples that are topologically distant but provide weak discriminative signals for contrastive learning. Conversely, an overly large $\theta$ (e.g., 8 or 10) excessively relaxes the constraint, inadvertently including items with moderate co-occurrences into the negative set. This introduces false negatives and significantly reduces the number of effective, high-quality negative samples. Empirical results indicate that $\theta=5$ perfectly balances this trade-off, ensuring the negative samples are both reliable enough to avoid noise and challenging enough to provide strong optimization gradients.

---

### Decision · Program_Chairs · 2026-04-30

**Decision:**

Accept (regular)

**Comment:**

This paper introduces CCLRec, a framework designed to bridge the gap between structural collaborative signals and LLM-derived semantic knowledge in graph-based recommendation systems. By utilizing a structure-semantic consensus mining strategy, the model identifies high-confidence positive and negative pairs to enhance contrastive learning.

The reviewers found the problem motivation highly relevant and the consensus-driven design intuitive and well-structured. The empirical results on public benchmarks demonstrate consistent improvements over strong baselines, which the committee views as a solid contribution to the community.

The author rebuttal successfully addressed several key questions, including clarifying the contrastive loss formulation, justifying the hard-intersection strategy, and providing ablation details for hyperparameter sensitivity. Most reviewers were satisfied with these clarifications and found the core idea interesting enough for presentation.

Last but not the least, the authors must explicitly detail the use of multimodal/visual features in the main text, correct the loss formulation, strictly unify terminology ("structure-semantic"), and include the promised limitations section regarding interaction sparsity and scalability.